

# Differences in the support needs of children with developmental disabilities among groups of medical and behavioral needs

Victor B. Arias, Virginia Aguayo, Miguel A. Verdugo and Antonio M. Amor

Institute on Community Integration, Faculty of Psychology, University of Salamanca, Salamanca, Spain

## ABSTRACT

**Background/Aims**. Medical and behavioral needs are relevant in organizing and providing support strategies that improve the quality of life for children, along with their families. The present study aims to examine the impact of medical and behavioral needs on the need for support of children with disabilities.

**Methods**. Health and education professionals were interviewed using the Supports Intensity Scale-Children's version to estimate the support needs of 911 children and adolescents (5–16 years) with an intellectual disability, including 55.32% with an additional disability. For data analysis, a model for measuring support needs was specified, consisting of seven support needs factors and three method factors. In estimating the model, four groups of medical and behavioral needs were considered. The factor scores' means of the groups were compared through $t$-tests.

**Results**. Medical and behavioral needs had an impact on overall support needs. Differences were found in all support domains for medical needs. The greatest influence of behavioral needs was found in the Social and School participation areas but was indistinguishable between the mild and moderate levels of needs.

**Conclusions**. Medical and behavioral needs greatly affect the need for support in a child's daily life, so they need to be considered a priority for support services. The importance of standardized assessments is emphasized to help develop support strategies.

Corresponding author
Virginia Aguayo, aguayo@usal.es

## INTRODUCTION

A person's need for support arises when the context demands a task that is beyond their abilities. Depending on their skills and the facilitators available in the environment, the intensity of support required will vary. The pattern and intensity of support that a person needs to participate in activities of daily living are called "support needs" (*Schalock et al., 2010*; *Thompson et al., 2009*). Understanding persons by their support needs, rather than by their impairments, is related to an ecological-contextual approach to disability and the adoption of a support model. This approach emphasizes the idea that everyone needs

support, and what differentiates some individuals from others is the intensity of support required to participate in activities of daily living (*Thompson et al., 2009*).

The support model is increasingly applied in the field of intellectual and developmental disabilities (IDD). Individualized support plans are developed considering a person's support needs profile, combined with their wishes and goals (*Schalock, 2018*; *Thompson et al., 2018*; *Walker et al., 2018*). Furthermore, aggregated data on the support needs profiles of many individuals are useful in improving the effectiveness and efficiency of organizational and state resource allocation (*Chou et al., 2013*; *Giné et al., 2014*; *van Loon et al., 2013*). The support model is also extending because it offers a way to integrate and organize support strategies aligned with desired quality of life outcomes, allowing for the evaluation of the suitability of undertaken actions (*Lombardi et al., 2016*; *Verdugo, 2018*).

The support model considers five elements in assessing the individual's support needs (*Thompson et al., 2004*): personal competence (defined as intellectual and adaptive behavioral skills), the number and complexity of environments in which the person participates, the number and complexity of activities, the presence of exceptional medical needs, and the presence of exceptional behavioral needs. Thus, one individual may have higher support needs than another because of additional medical and behavioral needs or the existence of restricted environments for participation (*van Gorp et al., 2019*; *van Timmeren et al., 2017*; *Zijlstra & Vlaskamp, 2005*). Furthermore, persons with medical and behavioral needs might require greater support than those with fewer limitations, including a wider range of resources to be deployed.

The presence of medical and behavioral problems in the severity of a disability has been well established, as it suggests important implications for the planning of support systems. Regarding the family, parenting becomes even more complicated, causing greater stress and the urgent need to care for the well-being of the individual and their family (*Kyzar et al., 2012*). Concerning school, teachers are required to take on the roles of health professionals along with their educational competencies, so they often work outside of their field of expertise (*Petitpierre et al., 2007*). Addressing extensive support needs is also a challenge for health systems because of the difficulty in identifying needs (*Bjorgaas, Hysing & Elgen, 2012*; *Mazza et al., 2019*) and mainly the lack of consistent terminology (*Nakken & Vlaskamp, 2007*), which hinders the allocation of services, especially in the case of severe disabilities.

From a support model, a person will need maximum levels of support given certain medical conditions and challenging behaviors, regardless of the intensity of the support needed in other areas of daily living (*Thompson et al., 2002*). Although the relationship seems clear, some authors (*Seo et al., 2017*) advocate examining the impact of medical and behavioral needs in different contexts to identify priorities in the provision of support. Surprisingly, this research is scarce, especially in children with IDD, and using a standardized measure of support needs. *Bertoncelli et al. (2019)* studied factors associated with severe intellectual disabilities in adolescents with cerebral palsy. They found that youths who require more medical support had more severe disabilities but did not require much behavioral support. It seems that support needs, especially relating to medical support, increase with the presence of other comorbidities and health problems, which is

common in more severe disabilities where there is a high co-occurrence of functional motor and intellectual impairments and limitations. Similarly, in a sample of 1,614 children with autism and intellectual disabilities, *Shogren et al. (2017)* found low medical needs and more behavioral needs. The authors related this finding to the increased risk of those behavioral needs in response to challenging contexts for children with autism. To our knowledge, no studies have analyzed the impact of medical and behavioral needs, considering different levels of intensity and assessing children with different IDD.

## Objectives of the study

The current study aims to analyze the impact of medical and behavioral needs on overall support needs in a sample of children with IDD. Consistent with other studies, we expect that a higher intensity of medical and behavioral support needs will lead to greater support needs overall. However, our goal is to investigate the nature of this relationship (i.e., whether it is linear or nonlinear) and to identify differences in the impact of exceptional needs across different domains of support. We also aim to analyze the effects of medical and behavioral needs on support need scores separately by considering different intensity groups (i.e., no needs, low needs, moderate needs, and high needs).

## MATERIALS & METHODS

### Participants

The voluntary collaboration of centers and entities specialized in IDD was requested. The selection criteria included the provision of support services to (a) children and adolescents between the ages of 5 and 16, and (b) having an IDD diagnosis.

The sample was composed of 911 children and adolescents (ages ranging from 5–16 years, $M = 11.15$, $SD = 3.42$), and predominantly male participants (61.91%). Most of the individuals lived with their parents (95.28%) and attended special education schools (67.18%). They belonged to 13 Spanish regions, including Castile and Leon (23.27%), Andalusia (15.70%), and the Community of Madrid (13.50%); their native language was Spanish (93.96%).

All participants had an intellectual disability, 44.24% of them as the primary diagnosis (Table 1). Intellectual disability was concomitant with cerebral palsy, autism, or sensory disability in 38.53% of cases, and three conditions were present in the 16.79% of children. Levels of intellectual functioning were collected from the children's medical records.

The information about the children's support needs was provided by direct-care professionals, comprised mostly of teachers (64.54%), psychologists (5.82%), social educators (5.05%), and physical therapists (4.72%).

### Instrument

To quantify and describe the profile of support needs, we used the Supports Intensity Scale-Children's version (SIS-C; *Thompson et al., 2016*) adapted to Spanish (*Verdugo et al., 2014*; *Verdugo et al., 2016*). The SIS-C is an objective tool that aims to assess the extraordinary support needs that youngsters (between 5 and 16 years old) with IDD need to participate in different activities of their daily lives successfully. The SIS-C is completed

**Table 1 Descriptive statistics for the children characteristics (n = 911).**

| Variables | n (%) |
|---|---|
| **Gender** | |
| Male | 564 (61.91%) |
| Female | 347 (38.09%) |
| **Age cohorts** | |
| 5–6 | 121 (13.28%) |
| 7–8 | 121 (13.28%) |
| 9–10 | 119 (13.06%) |
| 11–12 | 165 (18.11%) |
| 13–14 | 214 (23.49%) |
| 15–16 | 171 (18.77%) |
| **Home setting** | |
| Family home | 868 (95.28%) |
| Others (residential homes, etc.) | 35 (3.62%) |
| *Missing data* | 10 (1.10%) |
| **School setting** | |
| Ordinary school | 162 (17.78%) |
| Special classroom in ordinary school | 131 (14.38%) |
| Special education school | 612 (67.18%) |
| *Missing data* | 6 (0.66%) |
| **Health condition** | |
| Intellectual disability | 403 (44.24%) |
| Intellectual disability and cerebral palsy | 135 (14.82%) |
| Intellectual disability and autism | 176 (19.32%) |
| Intellectual disability and sensory limitations | 40 (4.38%) |
| Intellectual disability, cerebral palsy and sensory limitations | 145 (15.92%) |
| Intellectual disability, autism and sensory limitations | 8 (0.88%) |
| *Missing data* | 4 (0.44%) |
| **Estimation of limitations in intellectual functioning** | |
| Mild | 189 (20.75%) |
| Moderate | 302 (33.15%) |
| Severe | 259 (28.43%) |
| Profound | 124 (13.61%) |
| *Missing data* | 37 (4.06%) |

in the form of an interview by a qualified professional trained in its application, and its duration varies between 45 and 60 min. The respondents are individuals who have some sort of relationship with the child (e.g., direct care professionals or relatives), so their support needs can be estimated accurately.

The SIS-C is possibly the most widely tool used for standardized assessment of support needs (*American Association on Intellectual and Developmental Disabilities, 2020*). Its development emerged from a review of the scientific literature and Q-test methodology (*Thompson et al., 2002*; *Thompson et al., 2014*). The original scale analysis was performed
with 4,015 students, and good psychometric properties were obtained exceeding values of .90 in Cronbach's alpha (*Thompson et al., 2016*). Moreover, the structure of seven correlated support needs factors and three methods factors has been increasingly supported (*Aguayo et al., 2019*; *Seo et al., 2016*; *Seo et al., 2017*; *Verdugo, Arias & Guillén, 2019*). In all standardization studies, including those from other countries, trained professionals were responsible for collecting the participants' data (*Giné et al., 2017*; *Thompson et al., 2016*; *Verdugo et al., 2016*).

The SIS-C consists of two sections: (1) exceptional medical and behavioral needs and (2) support needs in different domains of the person's life.

### Exceptional medical and behavioral needs

This section is divided into two sets of activities, which are estimated on a scale of 0–2, with 0 being no support needed, 1 being some support needed, and 2 being extensive support needed. The rating in this section is not included in the SIS-C standardized scores, but it is taken as information that might influence the supports delivered in the support domains (*Thompson et al., 2016*). The set of medical needs consists of a total of 18 activities, which are related to respiratory care (e.g., postural drainage), feeding assistance (e.g., use of nasogastric tube), skin care (e.g., turning or positioning), or other forms of care (e.g., seizure management, dialysis). The set of behavioral needs includes 13 activities, either directed at individuals (e.g., prevention of injuries to others) or oneself (e.g., self-injury). Besides, these activities include inappropriate sexual behavior (e.g., sexual aggression) as well as other issues (e.g., preventing tantrums, wandering).

### Support needs in different domains

This section consists of 61 activities distributed in seven support domains, as follows: Home life (9 items; e.g., sleeping), Community and neighborhood (8 items; e.g., shopping), School participation (9 items; e.g., participating in activities in common school areas), School learning (9 items; e.g., learning academic skills), Health and safety (8 items; e.g., maintaining physical fitness), Social activities (9 items; e.g., making and keeping friends), and Advocacy (9 items; e.g., expressing preferences). The support for each activity is estimated through three indicators: the type of support needed, the frequency of support required, and the daily support time. The three indicators are scored separately and based on a Likert-type scale ranging from 0 to 4, where higher numerical values indicate the greater intensity of support. The seven-dimensional structure has been well established (*Seo et al., 2016*; *Verdugo et al., 2016*; *Verdugo, Arias & Guillén, 2019*).

## Procedure

This research was conducted under the principles of the Declaration of Helsinki (*World Medical Association, 2013*) and approved by the Bioethics Committee of the University of Salamanca. Schools, nursing homes, and early care centers providing support to children with IDD were contacted for data collection. Those interested in collaborating were sent more detailed information, and meetings were arranged to complete the SIS-C. Informed consent was signed by the parents or guardians of each participating child at the beginning of the study. Personal data were stored and protected, guaranteeing the confidentiality and

anonymity of the participants. The collaboration in this study was voluntary and free of charge.

The SIS-C was administered in pencil-and-paper format by a trained interviewer. In our research, about 70 interviewers, mainly psychologists, were trained to complete the scales. Based on our field notes, additional guidance was provided to them when the support needs of children with greater physical and communicative limitations were estimated, following the recommendations of *Schalock, Thompson & Tassé* (*2018*, p. 26). Once data collection ended, reports were returned, along with a certificate of participation. Telephone and email communication were constant between participants and the research team.

## Data analysis
### Missing data
Fourteen cases (1.5%) showed some missing data. Given the low prevalence and the fact that the coverage variable was higher than 99% in all cases, we used pairwise deletion to handle missing data.

### Specification of the SIS-C measurement model
According to *Seo et al. (2016)* and *Verdugo, Arias & Guillén (2019)*, we specified the SIS-C model from a multitrait-multimethod (MTMM) approach, in which support needs indicators were measured by seven support needs factors (corresponding to the seven domains) and three method factors (type, frequency, and time). Given several traits measured by different methods, the MTMM framework (*Campbell & Fiske, 1959*) allows for the assessment of convergent validity (i.e., that different assessment methods have concurrent validity in measuring the same construct) and discriminant validity (i.e., the degree to which different constructs measured with the same method are empirically separable). The MTMM approach has been integrated into confirmatory factor analysis (*Jöreskeg, 1971*; *Marsh & Hocevar, 1988*; *Widaman, 1985*) through the development of nested model taxonomies. Its usage is recommended when the measurement method is expected to affect the indicators under evaluation. In the case of the SIS-C, the effects of the method on support needs indicators reach 12% (*Aguayo et al., 2019*), so its incorporation to the model is necessary to obtain unbiased estimates on the latent variables of interest.

We grouped indicators (i.e., items) of each factor in parcels due to the complex parameterization of the SIS-C model (where a complete estimation of the raw data would result in a model with more than 1,000 parameters). The problem with over-parameterized models is how they tend to get a poor fit due to the cumulative effect of small errors of specification; in these cases, it is justified to reduce the number of parameters by creating parcels (*Marsh et al., 2014*; *Morin, Arens & Marsh, 2016*). We formed three parcels for each support factor, after verifying their correct functioning according to the recommendations of *Little et al. (2002)*. Each parcel was composed of the sum of the item scores for each measurement method (type, frequency, and time), following the instructions in the SIS-C manual for obtaining direct scores. Thus, Parcel 1 of each factor is the sum of the item scores considering the type of support, Parcel 2 is the sum of the scores to the frequency of support, and Parcel 3 is the sum of the scores given to daily support time.

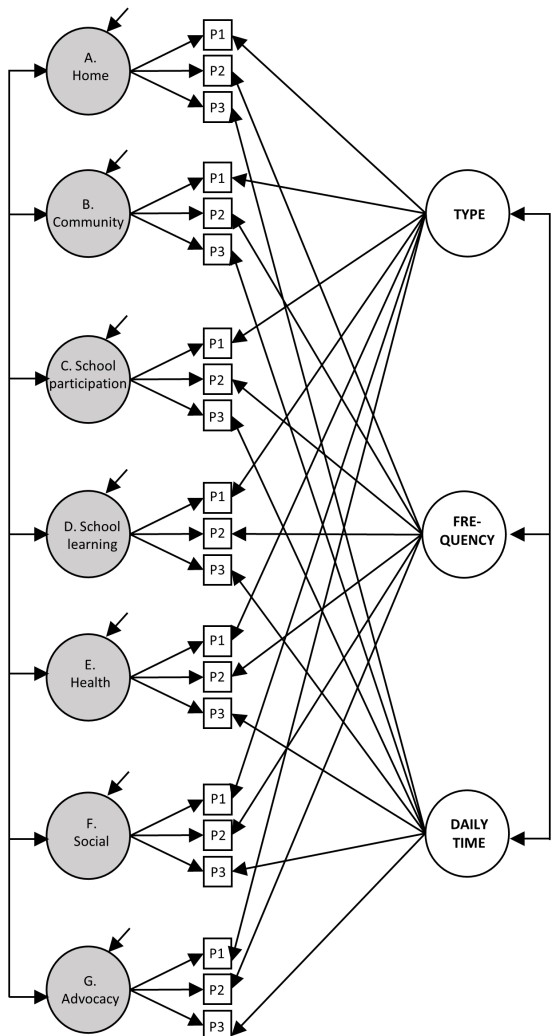

**Figure 1** **Representation of the measurement model of support needs.** The grey circles represent the seven factors of support needs, the white circles represent the three method factors, while the white squares represent the parcels (P). For clarity, the error terms of the parcels have not been represented.

The measurement model was specified, as shown in Fig. 1. The support needs factors comprised the sum of the scores of three parcels (e.g., the "Home" factor is measured by Parcels 1, 2, and 3 of that support factor). The method factors summed the parcels that referred to that method (e.g., the "Type of support" factor is measured by the Parcels 1 of all support factors). Correlations were also specified between the seven factors of support needs and between the three methods.

The model was estimated using robust maximum likelihood (MLR). The model fit was evaluated by considering the usual recommendations (*Browne & Cudeck, 1992*; *Hu & Bentler, 1999*): comparative fit index (CFI) and Tucker-Lewis index (TLI) above .90 and .95 indicate a good fit, while a root mean square error of approximation (RMSEA) index below .08 indicates an acceptable fit. The local fit was evaluated by examining the

modification indices (MI) in combination with the standardized expected parameters of change (SEPC). Significant MI values combined with SEPC values above .20 suggest the presence of possible relevant errors of specification (*Saris, Satorra & van der Veld, 2009*).

### Classification of individuals according to their medical and behavioral needs

The SIS-C manual recommends adding up the scores on the medical and behavioral needs items to obtain estimates of their intensity. However, we decided not to use the raw sum of scores since it is doubtful that items that refer to conditions of different severity (e.g., tube feeding and turning) will load similarly in the estimation of the needs. Instead, we used a formative model of measurement in which exceptional needs items were predictive of a principal component (*Borsboom, 2006*; *Preacher & MacCallum, 2003*). The resulting component was a weighted sum score that more accurately estimates the overall severity than unweighted sums of raw scores. Once the weighted scores were obtained, four groups were formed for each set of medical or behavioral needs: Group 1 (G1; no exceptional needs) included children who scored zero, and other groups included the remaining scores. Given the high positive asymmetry obtained, we classified in Group 2 (G2; low needs) the children with scores between the lowest fence and the median, in Group 3 (G3; moderate needs) the children with scores between the median and the upper quartile, and in Group 4 (G4; high needs) the children with scores between the upper quartile and the upper fence.

### Comparison between groups

Once a well-fitting measurement model of the SIS-C was obtained, we estimated the factorial scores of support needs (expected a posteriori method) and the weighted scores in the exceptional needs' components. From these scores, we estimated the differences in support needs between the four groups (G1, G2, G3, and G4) by $t$-tests for the difference of means. For its interpretation, we inspected the statistical significance, effect size, and factor score distributions in each group. Likewise, for a visual representation of the results, we elaborated figures with the factorial means and the confidence intervals (95%), following the recommendation of *Cumming & Finch (2005)*.

## RESULTS

### Specification of the SIS-C measurement model

The MTMM model had an acceptable fit at RMSEA (.061, CI= .056–.066) and a good fit at CFI (.98), TLI (.97), and SRMR (.01). The method factors captured 12% of the common variance, with the remainder (88%) assumed by the support needs factors. Moreover, the support needs factors explained 76.5% of the total variance of the data, indicating a good measurement quality. We then inspected the model modification rates, finding none extremely high to indicate any potential specification problems. Consequently, we decided not to introduce any modifications to the model.

### Classification of individuals according to their medical and behavioral needs

First, we estimated the optimal number of components to be extracted in the medical and behavioral needs sets by parallel analysis (*Horn, 1965*) on polychoric correlation matrices

with 1,000 permutations. In both sets, the analysis recommended retaining only one component, since the real data eigenvalues were lower than the average of the random eigenvalues from the second component onwards. Then, scores were estimated in each component, and the children were classified according to their level of exceptional needs. In terms of medical needs, 47.3% were classified in G1 (no needs), 27% in G2 (low needs), 13.4% in G3 (moderate needs), and 12.3% in G4 (high needs). In behavioral needs, 35.3% were classified in G1, 30% in G2, 20% in G3, and 14.7% in G4.

## Exceptional medical needs

Once the children were grouped according to the intensity of their medical needs, comparisons were made. Table 2 shows the results of $t$-tests for the difference of means, and Fig. 2 represents them in graphical format. We found significant differences ($p < .05$) among the four intensity groups in the seven support needs factors. The effect sizes ranged from 1.99 found in the Home factor between G1 and G4, to 0.28, in the Community factor between G3 and G4. Eventually following this pattern, the largest differences appeared for all support needs factors between G1 and G4 (large effect sizes, ranging from 1.09 to 1.99) and the smallest differences between G3 and G4 (small-medium effect sizes, ranging from 0.28 to 0.44). The distribution of groups suggested an asymptotic curve so that at lower intensities of support needs, there was a greater growth between groups than at higher intensities of support needs. This finding indicates that at a certain point (i.e., when medical needs are high), there was no relevant change in the profile of support needs in any area of daily living.

The higher the need for medical support, the greater the intensity of support needs. In fact, the increase between G1 (no needs) and G2 (low needs) was greater than 0.5 points, and the differences tended to be homogeneous in greater support needs. However, there were differences between the seven support needs factors. The largest differences among the groups were found in the Home and School participation factors, where the means differed by almost 1.5 points from each other, and effect sizes ranged from 0.45 1.99 to 0.34 1.5, respectively. Although the smallest differences were found in School learning (effect sizes between 0.33 and 1.08), the other support factors obtained similar values to this one.

The differences between the groups can be explored by looking at the error bars in the figures. The dispersion of scores was the most homogeneous in G4 of all the support needs factors, which is indicative of greater homogeneity in this group of needs. Likewise, the groups seemed more homogeneous in School learning and Health and safety.

## Exceptional behavioral needs

The differences between groups of behavioral needs (Fig. 3) showed a different pattern than that of the medical needs. First, all scores were found to be within one standard deviation (as opposed to almost two standard deviations in medical needs), suggesting less dispersion of scores and, therefore, less difference between the groups. Second, although significant differences were found in all support needs factors between G1, G3, and G4, this was not the case in the other comparisons. The differences between G1 and G2 were significant ($p < .05$) in all factors except Health and Advocacy; the differences between G2 and G4

**Table 2  Effect sizes of latent mean differences for medical and behavioral needs groups.**

|  | Medical support needs | | Behavioral support needs | |
|---|---|---|---|---|
|  | Effect size | Significance | Effect size | Significance |
| **Factor A. Home** | | | | |
| G1 vs. G2 | 0.81 | <.001 | 0.17 | .035 |
| G1 vs. G3 | 1.56 | <.001 | 0.26 | .005 |
| G1 vs. G4 | 1.99 | <.001 | 0.42 | <.001 |
| G2 vs. G3 | 0.77 | <.001 | 0.09 | .369 |
| G2 vs. G4 | 1.21 | <.001 | 0.26 | .013 |
| G3 vs. G4 | 0.45 | <.001 | 0.20 | .076 |
| **Factor B. Community** | | | | |
| G1 vs. G2 | 0.62 | <.001 | 0.23 | .005 |
| G1 vs. G3 | 1.13 | <.001 | 0.36 | <.001 |
| G1 vs. G4 | 1.36 | <.001 | 0.54 | <.001 |
| G2 vs. G3 | 0.57 | <.001 | 0.13 | .187 |
| G2 vs. G4 | 0.82 | <.001 | 0.32 | .003 |
| G3 vs. G4 | 0.28 | .033 | 0.24 | .039 |
| **Factor C. School participation** | | | | |
| G1 vs. G2 | 0.65 | <.001 | 0.24 | .004 |
| G1 vs. G3 | 1.21 | <.001 | 0.37 | <.001 |
| G1 vs. G4 | 1.50 | <.001 | 0.61 | <.001 |
| G2 vs. G3 | 0.62 | <.001 | 0.13 | .183 |
| G2 vs. G4 | 0.94 | <.001 | 0.39 | <.001 |
| G3 vs. G4 | 0.34 | .009 | 0.32 | .005 |
| **Factor D. School learning** | | | | |
| G1 vs. G2 | 0.46 | <.001 | 0.19 | .019 |
| G1 vs. G3 | 0.81 | <.001 | 0.26 | .005 |
| G1 vs. G4 | 1.08 | <.001 | 0.49 | <.001 |
| G2 vs. G3 | 0.39 | <.001 | 0.07 | .483 |
| G2 vs. G4 | 0.69 | <.001 | 0.31 | .004 |
| G3 vs. G4 | 0.33 | .012 | 0.27 | .018 |
| **Factor E. Health and safety** | | | | |
| G1 vs. G2 | 0.52 | <.001 | 0.12 | .132 |
| G1 vs. G3 | 0.92 | <.001 | 0.36 | <.001 |
| G1 vs. G4 | 1.18 | <.001 | 0.57 | <.001 |
| G2 vs. G3 | 0.45 | <.001 | 0.23 | .015 |
| G2 vs. G4 | 0.75 | <.001 | 0.45 | <.001 |
| G3 vs. G4 | 0.33 | .012 | 0.27 | .021 |

**Table 2** (*continued*)

| | Medical support needs | | Behavioral support needs | |
|---|---|---|---|---|
| | Effect size | Significance | Effect size | Significance |
| **Factor F. Social** | | | | |
| G1 vs. G2 | 0.53 | <.001 | 0.27 | .001 |
| G1 vs. G3 | 0.85 | <.001 | 0.55 | <.001 |
| G1 vs. G4 | 1.09 | <.001 | 0.83 | <.001 |
| G2 vs. G3 | 0.35 | .002 | 0.28 | .004 |
| G2 vs. G4 | 0.62 | <.001 | 0.58 | <.001 |
| G3 vs. G4 | 0.29 | .029 | 0.36 | .002 |
| **Factor G. Advocacy** | | | | |
| G1 vs. G2 | 0.46 | <.001 | 0.15 | .071 |
| G1 vs. G3 | 0.79 | <.001 | 0.35 | <.001 |
| G1 vs. G4 | 1.09 | <.001 | 0.55 | <.001 |
| G2 vs. G3 | 0.36 | .002 | 0.20 | .037 |
| G2 vs. G4 | 0.69 | <.001 | 0.42 | <.001 |
| G3 vs. G4 | 0.36 | .007 | 0.26 | .024 |

were significant in all cases; and the differences between G3 and G4 were significant in all cases except Home. However, the differences between G2 and G3 were significant only in Health, Social, and Advocacy. This result suggests that while the impact of behavioral support needs was progressive across intensity groups, there were fewer differences between medium intensity groups (i.e., G2 and G3).

Considering the support needs factors, those with the widest range were Social and School participation and that with the most homogeneity of scores was School learning. The largest effect sizes were found in Social (range 0.27 to 0.83) and School participation (range 0.13 to 0.61), and the smallest in School learning (range 0.07 to 0.49) and Home (range 0.09 to 0.42).

# DISCUSSION

## The present study

In this study, we compared four intensity groups of medical and behavioral needs to explore their influence on overall support needs in a sample of children with IDD. The results showed that medical conditions and challenging behaviors might cause extra support needs beyond the intensity of support needs in children with disabilities measured in the SIS-C. Differences were found in both medical and behavioral needs, suggesting their relevant effect on support needs.

For medical needs, analysis of the data gathered showed statistically significant differences in support needs among the groups in all seven support domains (as medical needs increased in intensity, so did the assessed support needs). The highest differences, as would be expected, were in comparisons between the G1 and G4 groups; the smallest (but still statistically significant) in the G3 and G4 comparisons. Effect sizes were generally consistent across domains, although the effect sizes in School learning domain comparisons were smaller than in the other domains. The finding that smaller differences were found in the

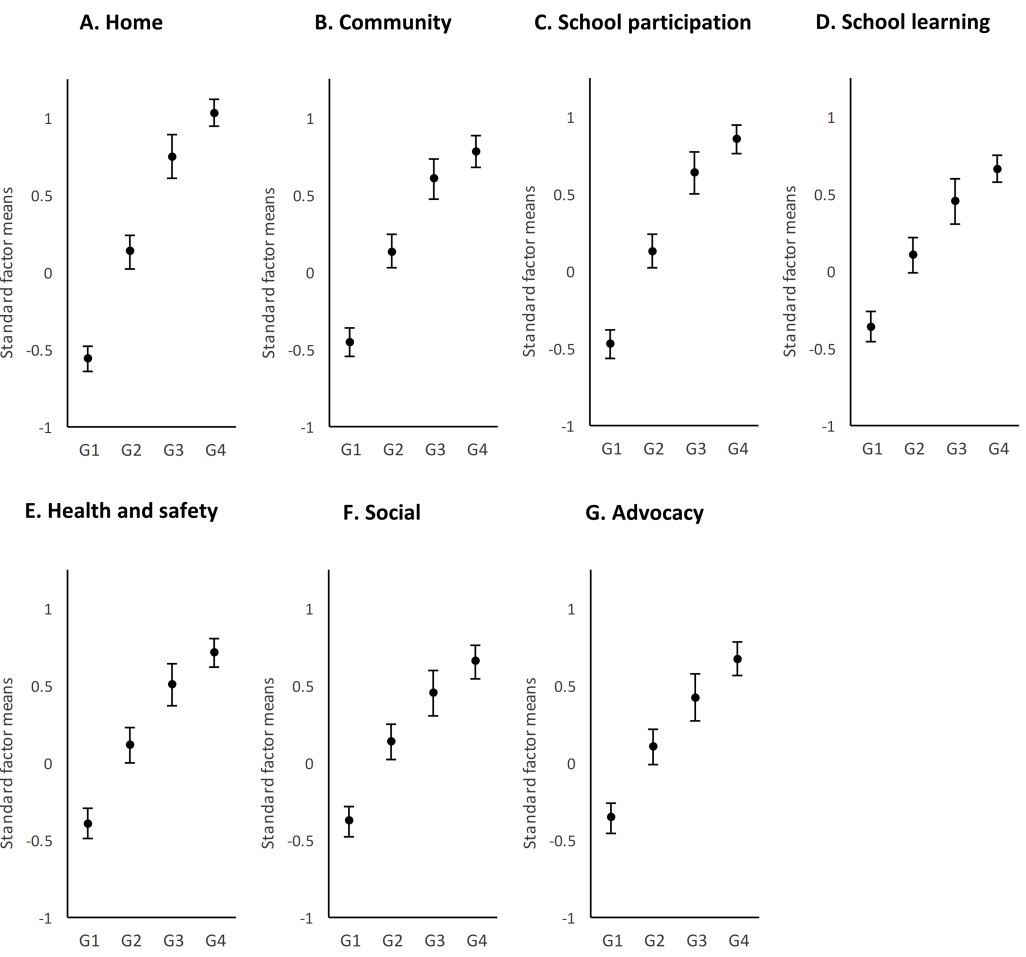

**Figure 2** **Distribution by support factors of the factorial means and the confidence intervals (95%) of the four groups of medical needs.** G1, no needs; G2, low needs; G3, moderate needs; and G4, high needs.

domain of School learning might be explained by considering that most of the participants attended special education schools and, therefore, received the common support of this educational context. Differences in the intensity of support needed for School learning may not be as accurately reflected in comparison with other domains, including School participation, where the support needs of students with intellectual disabilities are less dependent on the context than on the presence of special health care needs.

The impact of medical needs was non-linear but showed an increase in low levels of intensity (i.e., no needs, low needs) that tended to homogenize at higher levels (i.e., moderate and high needs). In turn, we found significant differences between having no needs and having any degree of needs, suggesting that medical needs have a considerable effect on the support needs of a child with IDD. This result is consistent with other studies, which conclude that the profile of support needs of individuals with severe and profound disabilities is different from those with mild impairments (*van Timmeren et al., 2017*). Support needs are maximal with severe disabilities, so there is little difference between

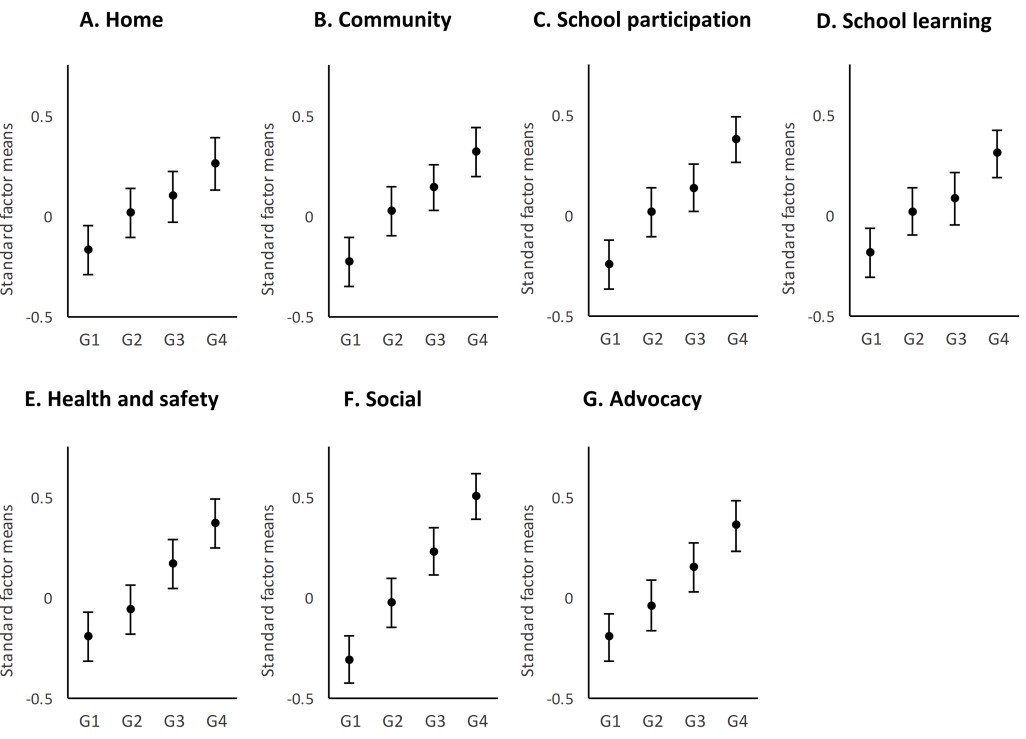

**Figure 3** **Distribution by support factors of the factorial means and the confidence intervals (95%) of the four groups of behavioral needs.** G1, no needs; G2, low needs; G3, moderate needs; and G4, high needs.

moderate and high levels of needs. This result depicts the expected increase in support needs given additional health problems, which is characteristic when several functional limitations occur together.

We found that the difference between having no behavioral needs and having high needs is significant ($p < .05$) in all areas. However, the influence of low-medium levels of needs appears to be dependent on the situation since it was significant only in some support domains. Thus, we identified that the greatest impact of behavioral needs was in the areas of Social and School participation, in which the relationship with peers is also the most common behavioral difficulty among children with higher impairments (*Parkes et al., 2008*; *van Gorp et al., 2019*).

The different impact of medical and behavioral needs has been shown in other research, where the medical needs were more related to support needs than behavioral ones (*Bertoncelli et al., 2019*). In turn, *Smit, Sabbe & Prinzie (2011)* suggested that the lower influence of behavioral needs could be attributed to the fact that the SIS's set of behaviors is mostly externalizing (e.g., aggression towards others) and, therefore, does not seem likely to be carried out by people with motor impairments. *Arnkelsson & Sigurdsson (2016)* found negative effects of the set of behavioral needs on overall support needs, which they attributed to an anomaly in the data. *Seo et al. (2017)* found that young adults with higher medical and behavioral needs had higher support needs, except in the areas of Community

and Health, where there were no significant differences with the group of high medical needs but rather low behavioral needs. They concluded that having high medical needs had more influence on those domains of support.

The stability of the need for support in persons with disabilities might explain our results. While medical support needs remain constant over time, behavioral needs are situational. Therefore, the difference between no behavioral needs and high needs is significant; however, at low and moderate levels, there is no impact on overall support needs, perhaps because some form of intermittent support should be provided anyway. Conversely, the intensity of medical needs increases enormously at low and moderate levels; however, it is no longer significant at high and very high levels of needs, where support is arranged on a permanent and ongoing basis. These results expose differences in medical and behavioral support planning that may be relevant to service eligibility and resource allocation (*Agosta et al., 2016*; *Fortune et al., 2009*). We also underpin that the problematic identification of behavioral or emotional problems in people with higher levels of support needs (*Bjorgaas, Hysing & Elgen, 2012*) might require sensitive items at all levels of intensity of support needs.

## Limitations

This study has some limitations that must be considered. Firstly, due to the sample size, we could not perform the analyses differentiating the results by health conditions (e.g., cerebral palsy or autism), as the variability within the groups was very small and did not allow us to obtain reliable analyses. While the type of health condition should not be relevant to a support needs assessment, which is primarily concerned with the individual's levels of functioning, considering different IDD may help to explore the effects of mobility or social behavior on the pattern of medical needs (as in the study on physical disability by *Smit, Sabbe & Prinzie, 2011*) or behavioral needs (as in the study on children with autism by *Shogren et al., 2017*).

Second, we did not analyze the contextual variables that could help us understand the support profile of the children assessed. An assessment of support needs should consider environmental aspects that may influence the support needs profile. In this sense, a study on access to health services or the possibility of participating in different activities in school would have been relevant.

Finally, a limitation of our study is that we did not use any other tool than SIS-C to collect data on such needs (medical and behavioral). For instance, we consider pain to be an important problem affecting the well-being of children with greater limitations and their performance in activities (*Horwood et al., 2019*; *Parkes et al., 2008*). Likewise, communication problems among children with greater limitations is a mediator in social and participation areas (*Caynes et al., 2019*), so it also may influence the need for support. Our results can only be interpreted by considering the SIS-C's sets of needs.

## Implications for practice and future research

The main implication of classifying individuals according to their support needs is resource allocation and the identification of common support strategies among individuals with

a similar level of needs. Support needs data also serves to select the priority areas of intervention. In this study, we found that the greatest need for medical and behavioral support principally influence the domains of Home, School Participation, and Social, being possibly the areas where more agents are involved, whether they are family, teachers, or peers and friends. Therefore, we join other authors' initiatives to propose support strategies for individuals with more significant needs, such as an improvement of the relationship with medical services, a family-centered care model, better opportunities for participation, along with enhancement of social inclusion and communication with peers (*Anaby et al., 2019*; *Ballard & Dymond, 2018*; *Finnerty, Jackson & Ostergren, 2019*; *Wehmeyer et al., 2016*).

Nevertheless, classifications based on support needs may be complicated by a lack of terminology to refer to children with significant support needs. In this regard, a standardized scale such as the SIS-C can be used. In our study, higher support needs have been characterized by a combination of medical and behavioral support needs affecting all areas of daily life. The SIS-C appears appropriate for assessing support needs, but further research is urged to address the evaluation of children with greater support needs.

## CONCLUSIONS

When there is a discrepancy between the personal competence and demands of the environment, support needs arise and guide the identification and provision of the best support strategies. We investigated the effect that medical and behavioral needs had on the support needs of children with IDD. Knowledge of the support needs of children is useful in informing work teams about priority areas that may be affecting the quality of their lives (*Petry, Maes & Vlaskamp, 2007*; *Schalock, 2018*; *Verdugo, 2018*). For guiding support planning, the usage of standardized tools for support needs measurement has been recommended.

## ACKNOWLEDGEMENTS

We are thankful for the effort and time of professionals and families who have voluntarily collaborated and participated in the present study.

### Funding
This work was supported by the Ministry of Economy, Industry, and Competitivity, Spain [PSI2012/36278 and PGC2018-093785-A-I00]. The funders had no role in study design, data collection and analysis, decision to publish, or preparation of the manuscript.

### Grant Disclosures
The following grant information was disclosed by the authors:
Ministry of Economy, Industry, and Competitivity, Spain: PSI2012/36278, PGC2018-093785-A-I00.

## Competing Interests

The authors declare there are no competing interests.

## Author Contributions

- Victor B. Arias and Virginia Aguayo conceived and designed the experiments, performed the experiments, analyzed the data, prepared figures and/or tables, authored or reviewed drafts of the paper, and approved the final draft.
- Miguel A. Verdugo conceived and designed the experiments, performed the experiments, authored or reviewed drafts of the paper, and approved the final draft.
- Antonio M. Amor performed the experiments, authored or reviewed drafts of the paper, and approved the final draft.

## Human Ethics

The following information was supplied relating to ethical approvals (i.e., approving body and any reference numbers):

The University of Salamanca granted Ethical approval to carry out the study within its facilities.

## Data Availability

The raw data are available as a Supplemental File.

## Supplemental Information

Supplemental information for this article can be found online at http://dx.doi.org/10.7717/peerj.9557#supplemental-information.

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
