# Peer review of "Differences in the support needs of children with developmental disabilities among groups of medical and behavioral needs"

_PeerJ, doi:10.7717/peerj.9557_

## Round 0.1 · original submission · Major Revisions

It looks like your paper will be publishable with suitable revisions.

·

Basic reporting

no comment (all is good); here are a few e minor text edits I suggest: on line 120 there should be no etc. after the e.g. - e.g., means example given and "etcetera" is not an example. Line 140 should be "Personal data were stored ..." not 'was" because data is a plural word.

Experimental design

Overall, the authors research procedures and data analyses were clearly communicated. I am confused about only one thing. The paragraph starting on line 178 refers to support needs factors and method factors. On line 202 the authors refer to substantive factors. Are the substantive factors the same as the support needs factors? if so, please use consistent terminology. If not, please clarify the differences.

Validity of the findings

no comment (the results made sense and were communicated clearly- nice job!)

Additional comments

I think an important omission in the discussion was that the authors did not comment on the potential influence that having the large majority of their participants attending a special school may of had on support needed for "school learning" activities. The SIS-C calls for rating support needs in relation to learning in general education classrooms, not learning in a special schools. School learning in a general education classroom for children with intellectual disability requires considerable support to differentiate instruction that promotes student learning compared to learning what is taught in a special school. On the other hand, supports that are needed for school participation may not be particularly different between the special school and general education classroom for students with intellectual disability and special health care needs. Participation is different from learning. My point is, since most of the students in this study were not in general education classrooms the intensity of supports needed for school learning may not be as accurately reflected compared to the other sub-scales, including school participation. Thus, the conclusion on line 254 may be partially an artifact of the context of the students' educational placement. I am not suggesting that the study was in any way flawed, but I am suggesting it is worth noting (with a sentence or two) that the finding that the influence of medical conditions on support needs was different in the school learning activities domain compared to the other support needs domains might be explained by the context in which the majority of participants were receiving their education. At least, it is a question for future researchers to investigate. I hope this critique is helpful.

Reviewer 2 ·

Basic reporting

Abstract: authors are suggested to add methods performed for the analysis of data (e.g.: four groups were compared through…)

Introduction/background: A greater attention should be given to the background and theory supporting this research, strengthening the specific importance of such study. Therefore, authors are suggested to deep in the medical and behavioral support needs’ impact in daily life/supports provision instead of just focused the support model (which is well done as a form to introduce the topic). This will help in discussion or in conclusion to add some practical recommendations for the identification of priorities (as stated in line 67). Further, some studies and results that identified factors, even if with other type of disability, could be detailed (e.g.: line 70-71: youths with severe intellectual disability need more medical support but not much behavioral support – do original authors give some explanation?).

Minor editings
Although I’m not an English native speaker and the writing is generally clear it would benefit from minor English editing (e.g.: line 91: The sample comprised of 911 children…)

Experimental design

Participants: maybe authors could add why children do not signed the informed consent or at least gave their oral consent; how authors controlled the “co-morbility associated” dependence? In participants section authors could add more information about the respondents (including the type of professionals/family member relationship, ages….
Instrument: I understand the tension between space and content, but it would be good to provide some brief information on (a) psychometric properties and b) systems of collecting data by the original authors
Procedures: Please clarify information on interviewer (or interviewers) as well as all application procedures: how many trained interviewers and their main characteristics; was anyone else present besides participants and researchers? were field notes made during and/or after interviews? what was an average duration of the interview? How was intellectual functioning level’ measured?

Validity of the findings

Discussion: this section has not, in my view, been expanded sufficiently beyond the descriptive. There needs to be more emphasis on why these results are important and the precise implications (e.g.: line 288: effect sizes in School learning domain were smaller than in other domains; little differences between moderate and high levels of needs – was this expected?; presence of medical need is no longer significant at high and very high levels ). Maybe authors could strengthen this section adding some explanations. Author(s) are also invited to strengthen the discussion: what is the worth-value of such a study and how findings could be “used” in others countries?
Finally authors are encouraged to add some recommendations based on this study findings.

Additional comments

Thank you for the opportunity to review this manuscript that presented research in an important area and provides more evidences in children with developmental disabilities’ characterization. Medical and behavioral needs and its assessment is relevant data for a more adjusted supports provision, and for a life with more quality. The sections are appropriate with suitable headings for a research article, the literature is well referenced and relevant and all data is supplied. It appears a carefully designed and rigorous study. Still some aspects of the manuscript could get improved and I will described some comments, hoping that my suggestions would further strengthen the manuscript.

---

## Round 0.2 · accepted · Accept

Thank you for making the changes suggested by the reviewers.